# Contributions of distinct gold species to catalytic reactivity for carbon monoxide oxidation

Li-Wen Guo[1,*], Pei-Pei Du[2,*], Xin-Pu Fu[1], Chao Ma[3], Jie Zeng[3], Rui Si[2], Yu-Ying Huang[2], Chun-Jiang Jia[1], Ya-Wen Zhang[4] & Chun-Hua Yan[4]

Small-size (<5 nm) gold nanostructures supported on reducible metal oxides have been widely investigated because of the unique catalytic properties they exhibit in diverse redox reactions. However, arguments about the nature of the gold active site have continued for two decades, due to the lack of comparable catalyst systems with specific gold species, as well as the scarcity of direct experimental evidence for the reaction mechanism under realistic working conditions. Here we report the determination of the contribution of single atoms, clusters and particles to the oxidation of carbon monoxide at room temperature, by the aid of *in situ* X-ray absorption fine structure analysis and *in situ* diffuse reflectance infrared Fourier transform spectroscopy. We find that the metallic gold component in clusters or particles plays a much more critical role as the active site than the cationic single-atom gold species for the room-temperature carbon monoxide oxidation reaction.

[1] Key Laboratory for Colloid and Interface Chemistry, Key Laboratory of Special Aggregated Materials, School of Chemistry and Chemical Engineering, Shandong University, Jinan 250100, China. [2] Shanghai Synchrotron Radiation Facility, Shanghai Institute of Applied Physics, Chinese Academy of Sciences, Shanghai 201204, China. [3] Hefei National Laboratory for Physical Sciences at the Microscale, University of Science and Technology of China, Hefei 230026, China. [4] Beijing National Laboratory for Molecular Sciences, State Key Lab of Rare Earth Materials Chemistry and Applications, PKU-HKU Joint Lab in Rare Earth Materials and Bioinorganic Chemistry, Peking University, Beijing 100871, China. * These authors contributed equally to this work. Correspondence and requests for materials should be addressed to C.M. (email: cma@ustc.edu.cn) or to R.S. (email: sirui@sinap.ac.cn) or to C.-J.J. (email: jiacj@sdu.edu.cn).

The extraordinary development of gold catalysis during the last two decades was initiated by the significant finding that gold nanoparticles strongly interacting with an oxide support are extremely active for the CO oxidation reaction[1,2]. Since then, numerous studies have confirmed the reactivity of nanosized Au species, as well as investigated the size effect of gold[3,4], the support effect of the oxide matrix[5,6], the nature of the active oxygen species[7,8] and the metal–support interactions[9,10]. However, there are still debates about the origin of the high reactivity of oxide-supported gold, even for the simple CO oxidation reaction. Among them, different active sites (single atoms[11], < 2 nm clusters[4,12] and 2–5 nm particles[13–15]) have been proposed previously. The difficulty is mainly due to the complexity of supported Au structures, resulting in mixtures of types of gold species in one sample, as well as the lack of effective structural characterization methods for the comprehensive detection of Au structures at the atomic level. Many research groups have dedicated themselves to distinguish the contribution of different Au structures (atoms[11,16], clusters[4,17–19], particles[13–15,20] and so on) with the help of new synthetic approaches. However, the preparation and stabilization of single-gold species on the same oxide matrix has not previously been achieved.

*In situ* techniques that emphasize the real catalyst, real working conditions and real-time detection play crucial roles in studies of the relationship between gold structures and catalytic performance. Although some efforts have been made using measurements such as high-resolution transmission electron microscopy (HRTEM)[4,11,14,21–23] and X-ray absorption fine structure (XAFS)[12,17,18,24–27], the precise *in situ* characterization of specific gold species, particularly single atoms, ultra-fine (< 2 nm) clusters and small-size (2–5 nm) particles, is still in great demand.

Here we report a facile synthesis of unique gold structures, including single atoms (Au_atom), < 2 nm clusters (Au_cluster) and 3–4 nm particles (Au_particle) anchored to CeO₂ nanorods. These gold-ceria catalysts show distinct reactivity for room-temperature CO oxidation. By the aid of *in situ* XAFS techniques, including X-ray absorption near-edge spectroscopy (XANES) and extended XAFS (EXAFS) analyses, we have successfully distinguished the catalytic contributions of the different gold species. Specifically, metallic Au species are much more important than ionic ones for the room-temperature CO oxidation, and the local coordination structure around gold in terms of the Au–O or Au–Au bonds is very stable during the reaction process. This work provides an opportunity to build a fundamental understanding of electronic and local coordination structure of the active site at the atomic level in the Au–CeO₂ system.

## Results

**Structural characteristics of gold catalysts.** From the synchrotron radiation X-ray diffraction patterns of single atoms (Au_atom), < 2 nm clusters (Au_cluster) and 3–4 nm particles (Au_particle) anchored to CeO₂ nanorods (Supplementary Fig. 1 and Supplementary Methods), we verified the pure Fluorite cubic CeO₂ phase for all the gold-ceria samples, except that a minor fraction of metallic gold was detected for Au_particle. The ceria structure was very stable throughout the *in situ* CO oxidation reaction (Supplementary Fig. 2), without any changes to the lattice constants of ceria. The corresponding X-ray photoelectron spectroscopy spectra confirmed that similar ratios of $Ce^{3+}/Ce^{4+}$ were present in the ceria nanorods for all the tested samples, whether as-prepared or used after the CO oxidation reaction (Supplementary Fig. 3 and Supplementary Methods). Thus, the

structural and textural properties of CeO₂ were identical for all three gold catalysts and they are suitable for the investigation of the role of different gold species in the catalysis.

The Au loadings were determined by inductively coupled plasma atomic emission spectroscopy as 1.2 and 0.8 wt% for Au_atom/Au_cluster and Au_particle, respectively. The HRTEM images show the rod-like shape of the ceria in the gold-ceria samples before (Supplementary Fig. 4a–c) and after the catalytic tests (Supplementary Fig. 4d–f). Creation of nanoparticles was only observed for Au_particle (Supplementary Fig. 4c,f) with an average size of $3.3 \pm 0.8$ and $4.1 \pm 0.8$ nm for the as-prepared and used samples (Supplementary Fig. 4g,h).

The aberration-corrected high-angle annular dark-field scanning transmission electron microscopy (HAADF-STEM) images clearly display the bright dots of atoms (Fig. 1a,b) and clusters (Fig. 1c,d), together with crystallized particles (Fig. 1e,f) for both as-prepared samples (Fig. 1a,c,e) and used catalysts after the CO oxidation reaction (Fig. 1b,d,f). The X-ray energy dispersive spectroscopy (EDS) spectra (Fig. 1g) taken from the yellow-box region show Au peaks, the intensity of which strongly depends on the aggregate size of the gold structures, that is, Au_particle > Au_cluster > Au_atom.

To identify the presence of single atoms in Fig. 1a,b, we took HAADF-STEM images with the ceria substrate off the zone axis, and bright dots of Au appeared near the edge of each nanorod. The simulated HAADF-STEM image is in good agreement with the experimental data (Fig. 2a–c). Specifically, for the simulated image in Fig. 2b, the substrate was tilted away from the zone axis and the thickness was set to 2.16 nm. It can be seen that the full width at half maximum of a single atom is 0.12 nm.

From Fig. 2d–f, we also found that the shape of Au_cluster changed over time under the high-intensity electron beam. So, we conducted the image collection in < 30 s in HAADF-STEM mode to avoid irradiation effects. All the measured samples remained stable with small dwell time (16 µs) for each pixel. Therefore, our analysis of different Au species in HAADF-STEM is reliable and can reveal the nature of the gold structures supported on the ceria nanorods. In our work, single atoms (Au_atom), < 2 nm clusters (Au_cluster) and 3–4 nm particles (Au_particle) were determined for the as-prepared samples and were very stable during the room-temperature CO oxidation reaction.

Owing to the high Z (atomic number) of Ce atoms, the contrast of Au species, especially for single atoms under the HAADF-STEM mode, may be indistinguishable from the thickness effect of ceria nanorods. Thus, we further applied XAFS to identify the local coordination structure of gold in each sample. The XANES spectra for the as-prepared gold-ceria samples in Fig. 3a reveal that ionic gold was dominant for Au_atom[27], while metallic Au was dominant for Au_cluster and Au_particle[27].

The EXAFS spectra in Fig. 3b, together with the corresponding fitting results (Supplementary Fig. 5 and Table 1), show the short-range (< 4 Å) local structure, including the distance ($R$) and coordination number (CN), of gold in the as-prepared samples. Two Au–O shells ($R = 1.96$ Å, CN = 3.0; $R = 3.49$ Å, CN = 5.7), without any Au–Ce contribution from Au–O–Ce structures, have been determined for Au_atom, indicating the presence of strong interactions between gold single atoms and surface oxygen species[27]. A pure Au–Au component ($R = 2.85$ Å, CN = 8.9) was confirmed for Au_particle, which corresponds to ca. 3 nm in average grain size[28]. The peak splitting in Fig. 3b is due to Ramsauer–Townsend resonance at a single energy in the backscattering amplitude of Au. Au_cluster has a much more disordered structure, exhibiting only 1/5 of EXAFS intensity as Au_atom or Au_particle. One Au–O shell ($R = 1.98$ Å, CN = 0.4) and one Au–Au shell ($R = 2.78$ Å, CN = 4.4, < 1 nm in averaged grain size) were present in Au_cluster. On the basis of the XANES

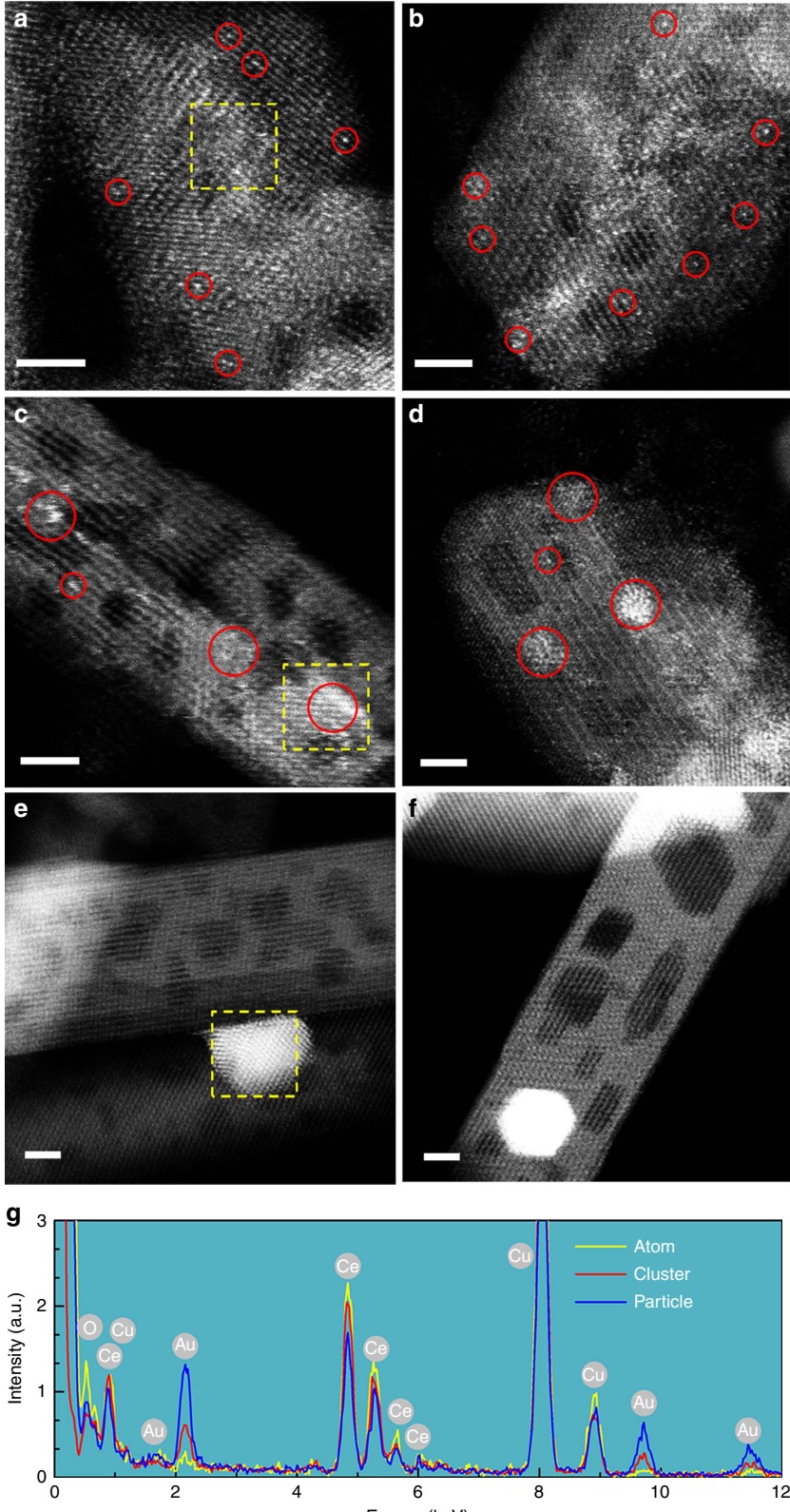

**Figure 1 | Aggregated size of gold structure.** HAADF-STEM images of the gold-ceria samples (**a,c,e**) before and (**b,d,f**) after CO oxidation: (**a,b**) Au_atom; (**c,d**) Au_cluster; (**e,f**) Au_particle; (**g**) EDS data collected from selected areas in **a,c** and **e**. Red circles: typical Au single atoms or clusters; yellow boxes: X-ray EDS regions. Scale bars, 2 nm.

and EXAFS results, a schematic of the gold structures in different samples is given in Fig. 3c.

According to the above experimental evidence from both HAADF-STEM and XANES/EXAFS, we have prepared three different single Au structures anchored onto the same type of ceria nanorods, and thus have good model catalysts to study the structure–function relations for gold catalysts. It should be noted that the single atoms of gold were stabilized at as high a

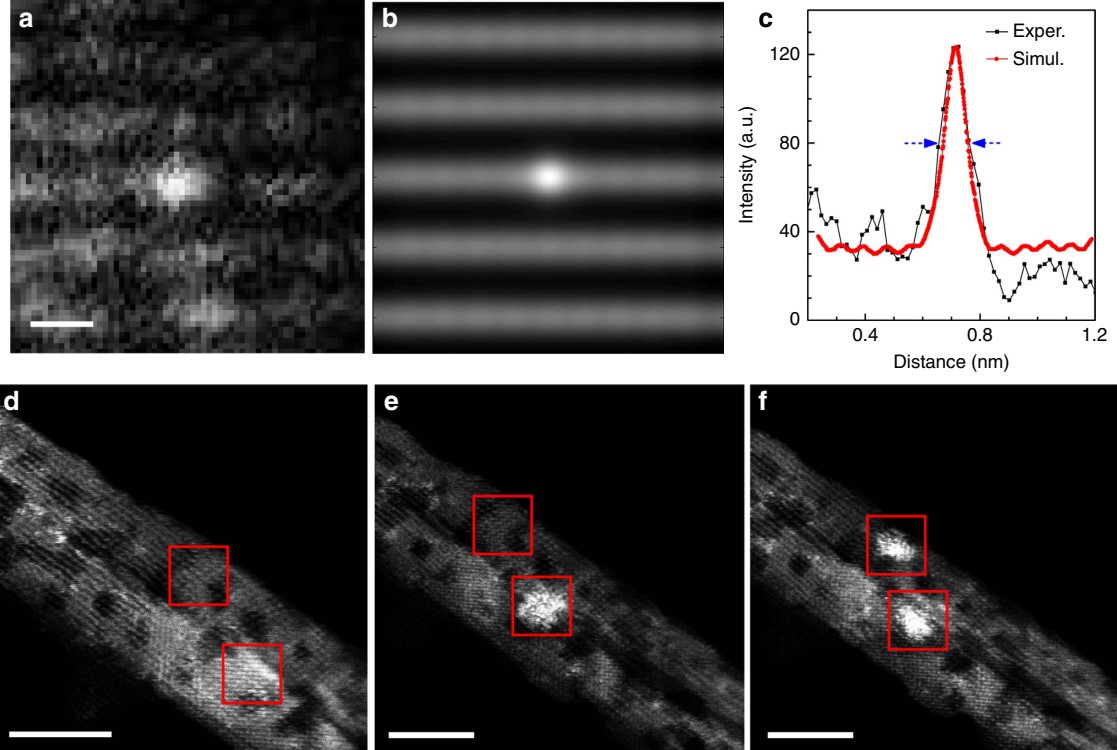

**Figure 2 | Cetification of Au single atom and shape transformation of Au cluster under electron beam.** (**a**) Experimental (Exper.) (**b**) simulated (Simul.) HAADF-STEM images and (**c**) the corresponding intensity profile for a typical Au single atom in the as-prepared Au_atom; scale bar in **a**, 0.2 nm. HAADF-STEM images from the as-prepared Au_cluster (**d**) before and (**e,f**) after electron beam irradiation over two different regions during the acquisition of X-ray EDS spectra (ca. 2 min). Red boxes: X-ray EDS regions; scale bars in **d–f**, 5 nm.

concentration as 1.2 wt% in our synthesis, probably due to the use of ceria nanorods as oxide support, since they can hold more oxygen vacancies on the {110} surface of $CeO_2$ (ref. 6).

**In situ exploration of gold species.** It has been argued for two decades that either ionic Au (refs 29,30) or metallic Au (refs 13,31) is catalytically active for the CO oxidation reaction. In this work, we tested the activity of the gold-ceria samples at room temperature. Figure 4a shows that the CO conversion rate per gram of gold follows the sequence of Au_particle > Au_cluster ≫ Au_atom. To monitor the structural evolution of the different gold species during the reaction, we carried out *in situ* XAFS measurements. First, X-ray photoelectron spectroscopy was performed to give the absolute gold valence in the as-prepared Au_atom/Au_particle samples (Supplementary Fig. 6), which were used as the ionic/metallic gold references. Then, a linear combination fit[32] was applied to all the *in situ* XANES profiles to determine the Au oxidation state at each step of the experiment (Supplementary Fig. 7). Meanwhile, the calculated ratio of $Au^{3+}/Au^{+}/Au^{0}$ as a function of reaction time was obtained (Fig. 4b). It can be clearly seen that the Au oxidation state was very stable without significant changes during the *in situ* tests. Finally, we found the relationship between gold oxidation state and the CO oxidation rate for the different gold species. Compared with the fully oxidized gold single atoms, the reduced gold clusters and particles have much higher activities towards room-temperature CO oxidation.

On the basis of the evidence above, we compared and analysed the related CO conversion rates (Fig. 4a), which were obtained from steady-state tests on Au_atom, Au_cluster and Au_particle (Supplementary Fig. 8). Obviously, gold was between the $Au^{+}$ and $Au^{3+}$ state in Au_atom, and the related reaction rate was in the range of 0–18 $\mu mol\,g_{Au}^{-1}\,s^{-1}$ (blue dots in Fig. 4a), while gold was between $Au^{0}$ and $Au^{+}$ state in Au_cluster, and the reaction rate was in the range of 253–281 $\mu mol\,g_{Au}^{-1}\,s^{-1}$ (green dots in Fig. 4a). For Au_particle, gold was very close to pure $Au^{0}$ and showed the highest rates of 392–467 $\mu mol\,g_{Au}^{-1}\,s^{-1}$ (red dots in Fig. 4a), which were more than 20-fold larger than gold in Au_atom. Considering the fraction of exposed Au sites (100% for single atoms and clusters and 30–40% for 3–4 nm particles), the CO conversion rate of Au_particle was over 100 times greater than Au_atom, normalized to the number of gold atoms on the surface. Furthermore, there was a slight increase (from 0 to 4%) in CO conversion for Au_atom with the reaction time (Supplementary Fig. 8), attributed to the partial reduction of the gold atoms $(Au^{+2.2} \rightarrow Au^{+2.0})$. Therefore, our *in situ* XANES results unambiguously demonstrated that metallic Au species play a more critical role in CO oxidation than their ionic counterparts in gold-ceria catalysts.

We also investigated the effect of the local Au structure on the reactivity of different gold species for the *in situ* XAFS measurements. By EXAFS fitting (Supplementary Fig. 9), we determined the R and CN values for each step of the experiment (Supplementary Table 1). Figure 4c shows that the CN of Au–O is nearly identical to be 3 for Au_atom, and the CN of Au–Au is nearly identical to be 9 for Au_particle. Adding the averaged EXAFS spectra for Au_cluster (Supplementary Table 1), we can demonstrate that all the fitted results from *in situ* XAFS were in good agreement with those for the as-prepared samples. Thus, such room-temperature reaction conditions were very mild, and no significant deformations were generated for our gold-ceria catalysts.

**Surficial adsorption.** To monitor the adsorbed species on the catalyst surface during CO oxidation, we further applied *in situ*

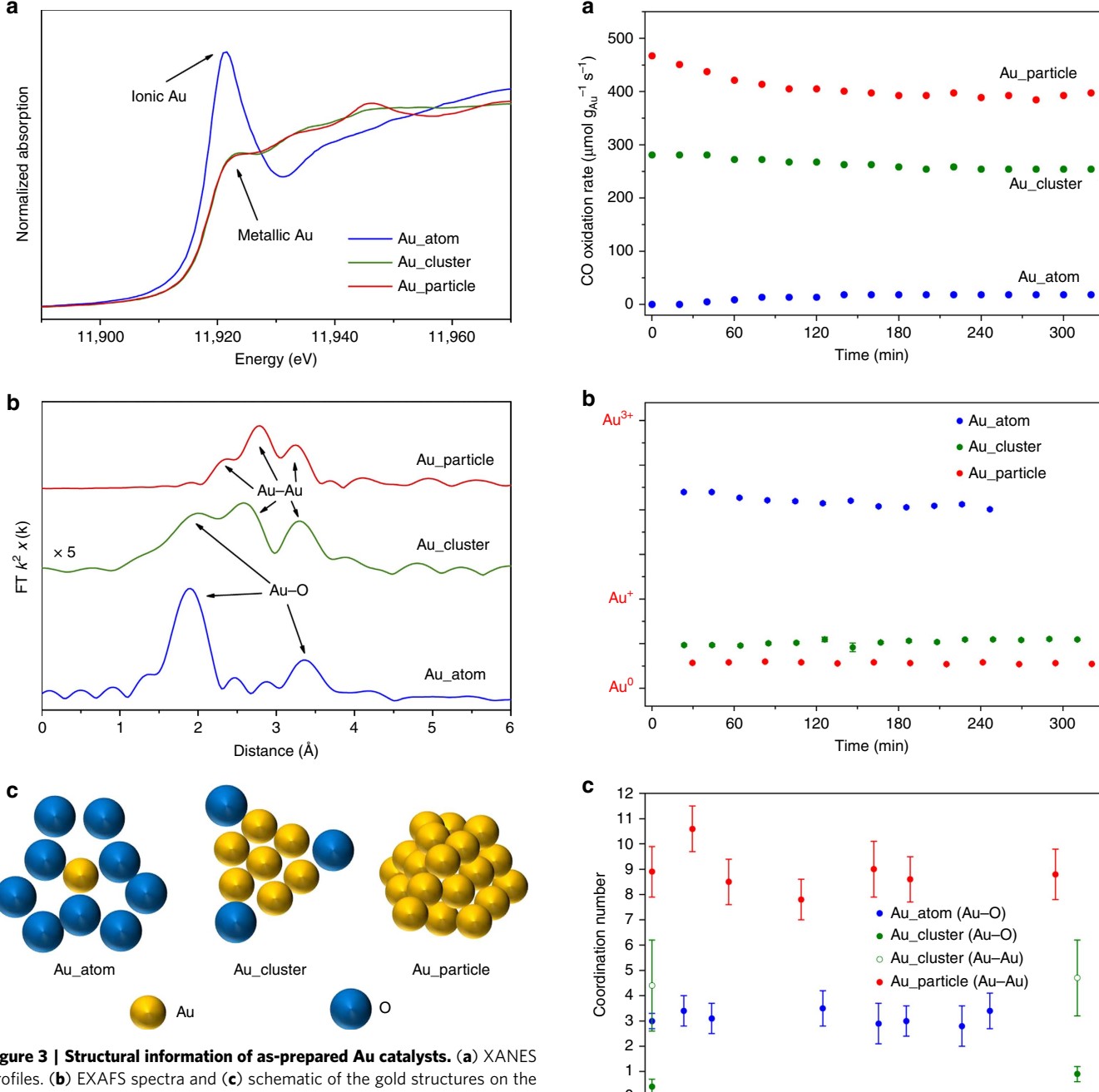

**Figure 3 | Structural information of as-prepared Au catalysts.** (**a**) XANES profiles. (**b**) EXAFS spectra and (**c**) schematic of the gold structures on the as-prepared gold-ceria samples.

**Figure 4 | Evolution of Au oxidation state and coordination number during CO oxidation conditions.** (**a**) CO conversion rate. (**b**) Au oxidation state and (**c**) coordination number of Au-O/Au-Au as a function of reaction time for the gold-ceria samples under the *in situ* CO oxidation conditions (25 mg catalyst, 1%CO/20%$O_2$/$N_2$, 20 ml min$^{-1}$, at 25 °C). The error bars in **c** are the fitting errors of coordination number, and have been calculated by EXAFS fittings.

**Table 1 | EXAFS fitting results on distance (R) and coordination number (CN) of Au-O and Au-Au shells for the as-prepared gold-ceria samples.**

| Sample | Au-O | | Au-Au | |
|---|---|---|---|---|
| | R (Å) | CN | R (Å) | CN |
| Au_atom | 1.96 ± 0.01 | 3.0 ± 0.3 | — | — |
| | 3.49 ± 0.03 | 5.7 ± 1.5 | | |
| Au_cluster | 1.98 ± 0.07 | 0.4 ± 0.3 | 2.78 ± 0.03 | 4.4 ± 1.8 |
| Au_particle | — | — | 2.85 ± 0.01 | 8.9 ± 1.0 |

diffuse reflectance infrared Fourier transform spectroscopy (DRIFTS) over the gold-ceria samples under steady-state conditions. The gas-phase $CO_2$ signals in DRIFTS at 2,340 and 2,360 cm$^{-1}$ (Supplementary Fig. 10) exactly follow the reactivity trends in the catalytic tests (Supplementary Fig. 8). For Au_atom, the Au$^+$–CO (ref. 29) band at 2,138 cm$^{-1}$, which overlapped with the gaseous CO signal, became more intense with reaction time (Supplementary Fig. 10a); for Au_cluster, the signal of Au$^{\delta+}$–CO $(0 < \delta < 1)$[30,33] at 2,123 cm$^{-1}$ was constant (Supplementary Fig. 10b), while for Au_particle, a continuous

decrease of the $Au^0$–CO (ref. 29) peak at 2,115 cm$^{-1}$ was observed (Supplementary Fig. 10c).

However, a severe overlap between gas-phase CO and chemically bonded Au–CO in the range of 2,169–2,118 cm$^{-1}$ appeared in DRIFTS under steady-state conditions. Thus, pulsed dynamic adsorption–desorption experiments (Supplementary Fig. 11) were conducted to distinguish the contributions of different gold species. Figure 5a shows the $Au^{3+}$–CO (refs 29,33 2,160 cm$^{-1}$)/$Au^+$–CO (ref. 29; 2,138 cm$^{-1}$), $Au^{\delta+}$–CO (refs 30,33; 2,124 cm$^{-1}$) and $Au^0$–CO (ref. 29; 2,115 cm$^{-1}$) bands for our gold-ceria samples. Typically, $Au^0$–CO was dominant for Au_particle and $Au^{\delta+}$–CO was distinct for Au_cluster, while $Au^+$–CO and $Au^{3+}$–CO were the species found in Au_atom. The differences in integrated intensity between CO adsorption and $O_2$ adsorption, that is, ability of CO adsorption, in pulsed DRIFTS experiments have been included in Fig. 5b. These results show a clear sequence of Au_particle > Au_cluster > Au_atom during 50 cycles or 100 switches. This matches the order of CO oxidation reactivity of the measured gold-ceria samples (Supplementary Fig. 8).

Furthermore, the CO titration results in Supplementary Fig. 12 indicate that both Au_cluster and Au_particle produced significant amounts of $CO_2$, corresponding to the product of CO molecules adsorbed by metallic gold and O atoms activated at the Au–$CeO_2$ interfaces (Supplementary Methods). However, Au_atom did not consume any CO at room temperature, due to either the weak adsorption of CO by ionic Au or the absence of activated oxygen species. These results are consistent with the *in situ* DRIFTS results.

## Discussion

In summary, we unambiguously determined the contribution of gold single atoms, clusters (< 2 nm) and particles (3–4 nm) for room-temperature CO oxidation under a non-interfering environment. With the help of controlled synthesis via wet chemistry, we obtained the above three pure gold species supported on the same type of ceria matrix. By the aid of *in situ* techniques, we traced the state of the Au species in the process of the reaction. The local structure around gold in the form of Au–O or Au–Au was very stable during the CO oxidation process. Particularly, compared with fully oxidized gold species in single atoms and the mixed cationic and metallic states in clusters, the highly reduced Au structure ($Au^0$) in particles has the strongest ability of CO adsorption and is much more critical as an active site for the CO oxidation reaction.

## Methods

**Preparation of ceria nanorods.** The ceria nanorods were synthesized according to the hydrothermal method[34]. Ce(NO$_3$)$_3$·6H$_2$O (4.5 mmol) was added into an aqueous NaOH (6 M, 60 ml) solution under vigorous stirring. After the precipitation process was completed (ca. 10 min), the stock solution was transferred into a Teflon bottle, and further tightly sealed in a stainless-steel autoclave. The hydrothermal procedure was carried out in a temperature-controlled electric oven at 100 °C for 24 h. The precipitates were separated by centrifugation and then washed with distilled water four times and ethanol once. The ceria support was obtained by drying the as-washed product in air at 70 °C overnight.

**Preparation of gold atoms supported on ceria nanorods.** The Au_atom sample was synthesized according to the deposition–precipitation method[6]. A 1 g sample of ceria support was suspended in 50 ml Millipore (18.25 MΩ) water. After stirring for 15 min, ammonium carbonate solution (25 ml, 1 M) was added. HAuCl$_4$·3H$_2$O (0.058 mmol) was dissolved in 25 ml Millipore water, and then added into the stock solution dropwise. After stirring and ageing at room temperature for 1 h, the as-formed precipitates were gathered by filtration and then washed with Millipore water at ca. 70 °C. The Au_atom sample was obtained after drying (70 °C, air, 4 h) and calcination (400 °C, air, 4 h).

**Preparation of gold clusters supported on ceria nanorods.** The Au_cluster sample was obtained by hydrogen treatment (5%H$_2$/He, 300 °C, 30 min) of the Au_atom sample.

**Preparation of gold particles supported on ceria nanorods.** The Au_particle sample was synthesized according to the colloidal deposition method[13], and poly(vinyl alcohol) (PVA, M$_w$ 10,000 from Aldrich, 80% hydrolysed) was used as the protecting agent. First, 0.675 ml of 0.5 wt% PVA solution (Au:PVA = 1.5:1 mg mg$^{-1}$) and 2 ml of 0.0125 mol l$^{-1}$ HAuCl$_4$ were added into 48 ml of Millipore water under vigorous stirring. After stirring for 10 min, a rapid injection of 1.30 ml of fresh 0.1 mol l$^{-1}$ NaBH$_4$ aqueous solution led to the formation of a wine-red solution. The support of 0.5 g ceria support was immediately added into the above colloidal gold solution under vigorous stirring. The completed adsorption of gold occurred after 0.5–1 h, which was indicated by the decolouration of the stock solution. The solids were gathered by filtration and washed with Millipore water to remove the dissolved impurities (for example, Cl$^-$). The Au_particle sample was obtained after drying (60 °C, air, overnight) and calcination (300 °C, air, 4 h). The containers were covered with aluminium foil to protect them from light.

**Transmission electron microscope.** HRTEM images, HAADF-STEM images and X-ray EDS were carried out on a JEOL ARM200F microscope equipped with probe-forming spherical-aberration corrector. Owing to the high Z (58) of the Ce atoms, the contrast of Au (Z = 79) in HAADF images is not clear, particularly for the thick region. Here, to enhance the contrast difference between the Au and Ce, we set the inner and outer angles of the HAADF detector to be 90 and 370 mrad, respectively, and the convergence angle to be ca. 30 mrad. The image simulation was performed using the multislice algorithm approach via the QSTEM software[35]. The spherical aberration coefficient was set to 0.5 μm.

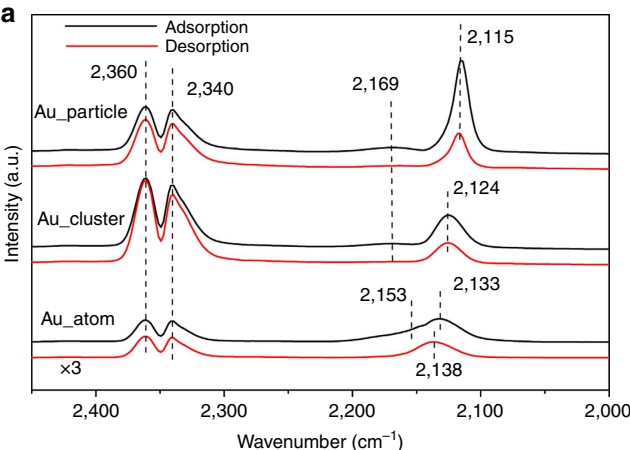

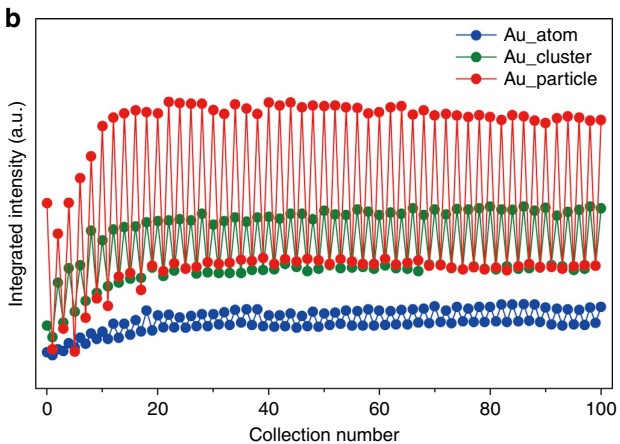

**Figure 5 | DRIFTS results of as-prepared Au catalysts.** (**a**) Typical *in situ* DRIFTS spectra from the pulsed adsorption–desorption experiment on the as-prepared gold-ceria samples (black: CO adsorption; red: CO desorption). (**b**) Integral intensity of CO bonded on metallic Au or ionic Au species as a function of reaction time during the pulsed adsorption–desorption experiments.

**X-ray absorption fine structure.** Before the *in situ* XAFS measurements, the Au L-III absorption edge ($E_0 = 11,919$ eV) XAFS spectra of gold-ceria samples were investigated *ex situ* at the BL14W1 beamline of the Shanghai Synchrotron Radiation Facility operated at 3.5 GeV under 'top-up' mode with current of 220 mA. The XAFS data were collected under fluorescence mode with a 32-element solid state detector. The *in situ* experiments were conducted at the X18B beamline of the National Synchrotron Light Source at Brookhaven National Laboratory, operated at 2.8 GeV under 'decay' mode with currents of $160 - 300$ mA. The powder sample (25 mg) was loaded into a Kapton tube (outer diameter = 0.125 inch), which was attached to an *in situ* flow cell. Two small-resistance heating wires were installed above and below the tube, and the temperature was monitored with a 0.5 mm chromel–alumel thermocouple that was placed inside the tube near the sample. The *in situ* CO oxidation reaction (1%CO/16%O$_2$/He, 20 ml min$^{-1}$) was carried out under a 'steady-state' mode at 25 °C. Before each reaction, the gold-ceria sample was pretreated in either 20%O$_2$/He (300 °C, 30 min) for Au_atom/Au_particle or 5%H$_2$/He (300 °C, 30 min) for Au_cluster. The gas composition in outlet was monitored by on-line mass spectroscope to confirm the start and end of the CO oxidation reaction. Each XAFS spectrum (ca. 15 min collection) was taken under fluorescence mode with a 4-element Vortex Silicon Drift Detector. The energy was calibrated for each scan with the first inflection point of the Au L-III edge in Au metal foil. Data processing and analysis were performed using the IFEFFIT package with Athena and Artemis software. The XANES linear combination analysis was done in $E$ space near the absorption edge (–20 to 30 eV). The EXAFS fittings were carried out by simulating the theoretical functions of Au calculated with FEFF6, as well as the Au–O first shell approach to the experimental data in $R$ space. The parameters describing the electronic properties (for example, correction to the photoelectron energy origin, $E_0$) and local structure environment (coordination number, bond distance and Debye Waller (DW) factor) around the absorbing atoms were allowed to vary during the fitting process. The corresponding $k$ weight in $k$ space was fixed to $k^2 -$ for the Au L-III edge.

**In situ DRIFTS.** *In situ* DRIFTS spectra were obtained using a Bruker Vertex 70 FTIR spectrometer fitted with an MCT detector. The DRIFTS cell (Harrick) was equipped with CaF$_2$ windows and a heating cartridge that allowed samples to be heated. An electro-control quick switching system was used to rapidly change the various gases with ultra-bit dead volume. About 30 mg of catalyst was calcined *in situ* in the reaction cell before the test (in a similar way as was used for the catalytic test: kept at 300 °C in air for 0.5 h for all samples; only for Au clusters, one extra following treatment in 5%H$_2$/Ar for 0.5 h was taken). Two types of DRIFTS tests were conducted at 25 °C in this work: steady-state mode and transient-pulse mode. For steady-state mode, the gas mixture (1%CO + 20%O$_2$ in N$_2$) was prepared using a mass flow controller with 30 ml min$^{-1}$ passing through the catalyst bed, which corresponds to conditions similar to the reaction tests in the tubular reactor. For the transient pulsed adsorption–desorption experiments, as in the scheme shown in Supplementary Fig. 11a, catalysts were tested for 50 cycles: the activated sample first had a CO adsorption by exposure to a 2%CO/N$_2$ gas mixture for 10 s, then rapidly switched to N$_2$, and the spectra were collected during the N$_2$-purging process. Then the catalyst had a CO adsorption again by exposure to a 2%CO/N$_2$ gas for 10 s and finally turned to 1%O$_2$/N$_2$ for 40 s with a simultaneous spectra collection. Typically, an acquisition time of 30 s at a resolution of 4 cm$^{-1}$ was used for spectrum collection. The intensities were evaluated in Kubelka–Munk units, and the background was obtained using a spectrum recorded in N$_2$ at the same temperature as above.

**Inductively coupled plasma atomic emission spectroscopy.** The inductively coupled plasma atomic emission spectroscopy analysis, which was carried out on an IRIS Intrepid II XSP instrument (Thermo Electron Corporation), was used to determine the gold/cerium concentrations.

**Catalyst test for CO oxidation.** The room temperature CO oxidation reaction on the gold-ceria samples was measured in a plug flow reactor by use of 25 mg of sieved (20–40 mesh) catalyst in a gas mixture of 1%CO/20%O$_2$/N$_2$ (99.997% purity) at a flow rate of 20 ml min$^{-1}$, corresponding to a space velocity of 48,000 ml h$^{-1}$ g$_{cat}^{-1}$. Before the measurement, for Au_atom and Au_particle, samples were pretreated in synthetic air (20%O$_2$/N$_2$) at 300 °C for 30 min, while for Au_cluster, the catalyst was *in situ* prepared from Au_atom through treatment in 5% H$_2$ in N$_2$ at 300 °C for 30 min, without further pretreatment. The catalytic tests were carried out in the above reactant atmosphere continuously at room temperature (ca. 25 °C) for over 6 h. The outlet gas compositions of CO and CO$_2$ were quantified online by non-dispersive infrared spectroscopy (Gasboard-3500, Wuhan Sifang Company, Wuhan, China).

**Data availability.** The data sets generated during and/or analysed during the current study are available from the corresponding author on reasonable request.

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

## Acknowledgements

Financial support was given from the National Science Foundation of China (NSFC) (grant nos. 21301107, 21373259, 21331001, 11079005 and 11405256), Fundamental research funding of Shandong University (grant no. 2014JC005), the Taishan Scholar project of Shandong Province (China), the Hundred Talents project of the Chinese Academy of Sciences, Open Funding from Beijing National Laboratory for Molecular Science and the Strategic Priority Research Program of the Chinese Academy of Sciences (grant no. XDA09030102). Thanks are due to Nebojsa Marinkovic (Delaware University) for his kind help on *in situ* XAFS measurements, and Zong-Yuan Liu and Wen-Qian Xu (Brookhaven National Laboratory) for *in situ* X-ray diffraction set-ups.

## Author contributions

R.S. and C.-J.J. supervised the work; L.-W.G., P.-P.D., R.S. and C.-J.J. had the idea for and designed the experiments, analysed the results and wrote the manuscript; L.-W.G. performed the catalyst preparation and catalytic tests; X.-P.F. and C.-J.J. performed the *in situ* DRIFTS measurements; P.-P.D., R.S. and Y.-Y.H. performed the XAFS measurements and analysed the results; C.M. and J.Z. performed the aberration-corrected HRTEM and STEM-EDS measurements and analysed the results; Y.-W.Z. and C.-H.Y. conducted the X-ray photoelectron spectroscopy tests; J.Z., Y.-W.Z. and C.-H.Y. gave helpful suggestions and fruitful discussion on the experiments design, data analysis and manuscript preparation.

## Additional information

**Competing financial interests:** The authors declare no competing financial interests.

**Publisher's note**: 

