## [Peer Review File · Nature Communications]

Reviewers' comments:

Reviewer #1 (Remarks to the Author):

The work submitted for publication in Nature Communications reports the synthesis and detailed characterization of three Au/CeO₂ samples containing atoms, small clusters and larger nanoparticles, and their catalytic testing in the CO oxidation reaction. As the authors explain, it is really difficult to prepare Au/CeO₂ materials with only one type of gold species. And similarly difficult to unambiguously characterize these materials. In this sense, the combination of HAADF-STEM and EXAFS-XANES techniques used in this work seems to provide characterization of the gold species present in each catalyst, and confirms the correct preparation of samples containing very homogeneous distribution of gold species.

Then, the catalytic activity of the three samples in the oxidation of CO has been tested, and it has been found that isolated atoms are not active, and that the largest particles containing metallic gold show the highest activity. With the assistance of in situ DRIFTS spectroscopy, the higher activity of Au particles has been related to a better adsorption of CO at these metallic sites. This analysis of data is perhaps too simple, and does not include the issue of where activation of O₂ occurs. There are many mechanistic studies about the CO oxidation reaction catalysed by gold showing that the activation of molecular oxygen preferentially occurs at the metal-support interface, involving or generating in the process partially cationic Au sites. But this does not mean that cationic gold is the catalytically active species. In fact, it has also been shown that isolated cationic gold atoms (like those present in the Au_atom sample described here) are not able to activate molecular O₂ for oxidation reactions. In my opinion, the discussion about the origin of the higher reactivity of the larger particles (3-4 nm) as compared to clusters (1-2 nm) and atoms should consider the activation of molecular oxygen, and the importance of each step (O₂ activation, CO adsorption, CO₂ formation) in the mechanism, in order to provide a more relevant chemical insight into the process. In conclusion, I recommend the authors to also consider the activation of molecular oxygen and the oxidation step, to provide a clear vision of the active sites involved. As it is, I consider the manuscript limited and not suitable for publication in Nat. Comm.

Reviewer #2 (Remarks to the Author):

A. This manuscript details the study of different gold samples (ionic salts, clusters and nanoparticles) on ceria for the oxidation of Carbon monoxide using in situ X-ray absorption spectroscopy and STEM to understand speciation of the Au catalysts. Results show that metallic gold species are present in samples having high catalytic activity. DRIFTS spectroscopy was also used to probe the nature of the Au species for each of the samples.

B. This is an interesting paper that has, for the most part, well validated conclusions from experimental work. The X-ray absorption work is very well done. As the authors note in the manuscript well, there have been a good number of other attempts to understand Au oxidation catalysts via X-ray absorption spectroscopy, and it is fair to say that those results have been all over the map in terms of identifying the catalytically active species. This is, in part, due to the difficulty in making samples that have distinctive single phases, but is also likely due both to X-ray beam damage of samples during acquisition and the fact that X-ray absorption spectroscopy is a bulk technique that is not well suited for identifying minor species. Nevertheless, I think the authors have done well to make three distinctive samples, and have commented on how they avoided beam damage in the samples, thus I think there is novelty in the final conclusions of this work, and it should be acceptable for publication in Nature Communications pending minor revisions.

C. Quality of data looks excellent for X-ray absorption work. It would be useful if the authors could provide higher magnification images of the STEM images in Figure 1, particularly for the atomic gold samples.

D. Statistical treatment of X-ray absorption work is very good, and uncertainties are reported.

E. The conclusions of the paper are well validated from experimental results.

F. Suggested improvements:

1. Improve magnification of STEM images (see point C).
2. Improve grammar throughout - currently the grammar in the paper is quite poor and detracts from the manuscript substantially.

G. References are appropriate, and I think previous X-ray absorption work on such systems is fairly well cited.

H. Clarity needs work - as mentioned in section F, the grammar in the manuscript submitted is quite weak, and could be improved significantly. This would greatly improve the readability of the paper.

Reviewer #3 (Remarks to the Author):

Gold nanocatalyst is famous for its extraordinarily high activity for low temperature CO oxidation. However, the nature of active sites has long been disputed, for example, cationic or metallic, single atoms or clusters or nanoparticles. The present work by Prof Jia and co-workers appears to reveal the tip of the iceberg. They prepared gold single atoms, clusters (<2 nm), and nanoparticles (3-4 nm) on a ceria nanorod support, and compared their activities for RT CO oxidation meanwhile detected the local structure change and oxidation state change of different gold species during the reaction. Based on these in-situ characterization results, they conclude that gold metallic nanoparticles are much more active than cationic gold single atoms for RT CO oxidation. I enjoyed the application of the in situ XAFS and dynamic adsorption-desorption IR techniques, which provide reliable evidence to support the conclusion. Nevertheless, the catalysis by gold is critically correlated with the support nature, which is not addressed in the work.

1. The authors claim that the three catalysts have the same support. However, the difference in preparation history of the catalysts may cause the change of the support surface properties. For example, single atom catalyst was calcined at 400 oC while the cluster catalyst was further reduced at 300 oC. The reduction treatment will bring about the change of surface lattice oxygen of CeO₂ support in addition to the aggregation of gold single atoms. CO titration in combination of XPS may help to address this concern.

2. The HAADF-STEM images seem to show that the ceria nanorods have a large number of voids, which means that it has defect-rich surface. Do these defects affect the catalysis by gold?

3. It is difficult to understand the models in Fig. 3c. Where do the oxygen atoms come from? The authors show two Au-O shells in single-atom catalyst but they claim that no any Au-Ce distance from Au-O-Ce structure. Does this mean gold single atoms do not coordinate with ceria via oxygen atoms? If it is true, how is it stabilized by the support?

4. In gold single-atom catalyst, 1.2 wt% Au loading is relatively high for atomically dispersion. I guess the ceria nanorod support has many defects that can stabilize single atoms of gold, is it right?

Responses to the reviewer's comments

Reviewer #1:

Comment 1: The work submitted for publication in Nature Communications reports the synthesis and detailed characterization of three Au/CeO₂ samples containing atoms, small clusters and larger nanoparticles, and their catalytic testing in the CO oxidation reaction. As the authors explain, it is really difficult to prepare Au/CeO₂ materials with only one type of gold species. And similarly difficult to unambiguously characterize these materials. In this sense, the combination of HAADF-STEM and EXAFS-XANES techniques used in this work seems to provide characterization of the gold species present in each catalyst, and confirms the correct preparation of samples containing very homogeneous distribution of gold species. Then, the catalytic activity of the three samples in the oxidation of CO has been tested, and it has been found that isolated atoms are not active, and that the largest particles containing metallic gold show the highest activity. With the assistance of in situ DRIFTS spectroscopy, the higher activity of Au particles has been related to a better adsorption of CO at these metallic sites. This analysis of data is perhaps too simple, and does not include the issue of where activation of O₂ occurs. There are many mechanistic studies about the CO oxidation reaction catalyzed by gold showing that the activation of molecular oxygen preferentially occurs at the metal-support interface, involving or generating in the process partially cationic Au sites. But this does not mean that cationic gold is the catalytically active species. In fact, it has also been shown that isolated cationic gold atoms (like those present in the Au_atom sample described here) are not able to activate molecular O₂ for oxidation reactions. In my opinion, the discussion about the origin of the higher reactivity of the larger particles (3-4 nm) as compared to clusters (1-2 nm) and atoms should consider the activation of molecular oxygen, and the importance of each step (O₂ activation, CO adsorption, CO₂ formation) in the mechanism, in order to provide a more relevant chemical insight into the process. In conclusion, I recommend the authors to also consider the activation of molecular oxygen and the oxidation step, to provide a clear vision of the active sites involved. As it is, I consider the manuscript limited and not suitable for publication in Nat. Comm.

Response: Thanks for the reviewer's comment. Activation of molecular oxygen truly plays the very important role for the CO oxidation catalyzed by supported gold catalyst, and recently the corresponding investigation has been received much attention.¹⁻⁸ Many remarkable processes have been achieved, such as that by the temporal analysis of products (TAP) technique, Behm and his coworkers have identified the oxygen atoms at the perimeter sites as the active species for the reducible metal oxide matrix in the supported Au nanoparticle system.⁵ In another system of Au/CeO₂, using isotopic switching technique, Overbury et al. demonstrated that in low-temperature CO oxidation, CO reacts with lattice oxygen of CeO₂, instead of directly with gaseous O₂.⁸ However, the exact mechanism on the molecular oxygen activation in heterogeneous catalysis is still unclear, much work is to be done. Besides the activation of molecular oxygen, the adsorption of the carbon monoxide is also crucial to the reactivity in CO oxidation;⁹ therefore we must consider these two factors together.

Based on the reviewer's suggestion, in our supplementary work, we have carried out some related investigations on the molecular oxygen activation over different gold catalysts. Firstly, the CO titration test was used to investigate the initial surface reaction of all gold catalysts and detect the possible active oxygen species in the surface of the Au/CeO₂ catalyst. As shown in Figure S12 (Figure L1 here), the pure ceria rods and Au_atom sample revealed no response of CO₂ formation ($m/z = 44$). However there are obvious evolutions of the formation CO₂ for the Au_cluster and Au_particle sample. From the CO titration results, we can clearly see that there are some active oxygen species (O_{act}) in the surface of the Au_cluster and Au_particle catalysts due to the formation of CO₂ that is from the reaction of CO with O_{act} . However, we cannot judge why there is no evolution of CO₂ for Au_atom catalyst because either the lack of O_{act} or the very weak adsorption of CO on Au_atom catalyst could be the reason. Focusing on this point, we have further conducted the related investigation on the CO adsorption using *in-situ* DRIFT technique under similar condition to that of CO titration test to monitor the status of the CO molecular in the reaction.

In-situ infrared spectroscopy has been deeply utilized in detecting surficial absorbance of CO molecule. Herein, the CO adsorption tests were conducted and the results are shown in

Figure L2. We found that for Au_atom catalyst, the signal of adsorbed CO (CO-Au) was covered by that of the gaseous CO (2170 and 2119 cm^{-1}) and hardly detectable. Conversely, the adsorption of CO in Au_cluster and Au_particle were evidently banded at 2115 cm^{-1} ,¹⁰ which is assigned to CO-Au⁰ species. Here, a slight red shift (9 cm^{-1}) occurred for the adsorbed CO in Au_cluster comparing with the pulse measurement and operando test shown in Figure 5 and Figure S11, which is owing to the presence of additional oxygen in pulse or operando test that makes the gold species more positive. It is clearly observed from the CO absorbance that the CO adsorbed abilities of various gold species were especially distinct with the following order: CO-Au_particle > CO-Au_cluster >> CO-Au_atom, which is well consistent with the CO oxidation reactivity of these catalysts at room temperature. It is noticed that in the region of 2300–2400 cm^{-1} , the signals of gases CO₂¹¹ over different catalysts give the same tendency to those got from CO titration tests. Based on this, we speculate that the adsorbed CO reacts with the lattice oxygen of CeO₂ via an MvK mechanism at room temperature, as proved by Overbury et al. by isotopic exchange experiments previously.⁸ Here we found there is obvious distinction in CO adsorption for different gold catalyst of Au_particle and Au_cluster and Au_atom, which makes it difficult to distinguish the respective contribution of O₂ activation and CO adsorption to the reactivity in CO oxidation.

Due to the above problem, finally, we used temperature programmed desorption of oxygen (O₂-TPD) technique. As valuable aid for detecting oxygen species of catalyst, O₂-TPD has been widely used in previous reports.^{2,12} By this means, Haruta and coworker observed the adsorbed behavior of oxygen species at low temperature over Au/Co₃O₄ catalyst, proving the role of O_{ad} during low-temperature CO oxidation.² On the other hand, Vayanas et al. have proved that the lattice oxygen played an important role in the oxidation reaction by the corresponding O₂-TPD spectra.¹² For the as-prepared gold catalysts in this work, as shown in Figure L3 on the O₂-TPD results, no O₂ signals were detected up to 800 °C for all the measured gold-ceria samples, which verifies the negligible quantity of absorbed oxygen at room temperature for either active (Au_particle and Au_cluster) or inactive (Au_atom) catalysts, which is consistent with some previous reports of Au/CeO₂ catalysts.^{8,13}

Additionally, the *in-situ* XRD results in our work (Figure S2) have shown no detectable distinction for all the supported gold catalysts, indicating that during the room temperature CO oxidation process, the reducible matrix were stable for all, which is also confirmed by the XPS results on the status of Ce species in Figure S3 (Figure L4 here). Therefore, we think there is very little difference on the molecular oxygen activation in this supported Au/CeO₂ (rod) system, which might be owing to the very low loading Au species and the stable structure of the same support materials (CeO₂ rods).

In summary, just as the reviewer's comment, it is very important of each step (CO adsorption, O₂ activation and CO₂ formation) for the CO oxidation based on the abundant previous studies. However, up to now, there are still many difficulties to get one fully unique image on the mechanism of CO oxidation reaction. **In our work, rather than the investigation of the mechanism of CO oxidation reaction itself, we focus more on the real active sites of gold catalysts in one comparable system. By comparing the features among Au_atom, Au_cluster and Au_particle supported on the same ceria nanorods, we try to figure out the relation between the various catalytic reactivities and the different Au species.** Following the reviewer's valuable advices, we have tried to further get more information the active surficial oxygen species. However, we cannot make a conclusion if there is distinction on molecular oxygen activation among these three different Au/CeO₂ catalysts of Au_atom, Au_cluster and Au_particle. Thus, we can only conclude that the metallic gold species play one more important role than cationic gold species in catalyzing CO oxidation reaction at room temperature which is related to the effective adsorption of CO on the gold.

Figure L1. CO₂ ($m/z = 44$) evolution collected during CO titration over gold-ceria samples at room temperature.

Figure L2. DRIFT spectra (2000 – 2500 cm⁻¹) of different gold-ceria samples: (a) Au_atom; (b) Au_cluster; (c) Au_particle. Data were collected on sample powders (ca. 20 mg) under the CO adsorption conditions (2%CO/He, 30 mL/min) at room temperature within the initial 10 min.

Figure L3. O₂ evolution ($m/z = 32$) during O₂-TPD over gold-ceria samples. The O₂-TPD experiments were performed at Builder PCSA-1000 System equipped with a mass spectrometer (AMETEK, DYCOR LC-D200). First, 100 mg sample powders were pretreated at 300 °C (10 °C/min) in air (50 mL/min) for 30 min and in addition subsequent pretreatment (300 °C, 5% H₂/He, 30 min) was required for Au_cluster. After cooling down, the measured sample was continued to be purged with pure O₂ (50 mL/min) at room temperature for 1 h to obtain saturated adsorption. Then, the feed gas was switched to pure He (30 mL/min) at room temperature until the stabilization of baseline. The O₂-TPD process was done in pure He (30 mL/min) from room temperature to 800 °C (10 °C/min) with the simultaneous collection on O₂ signals ($m/z = 32$).

Figure L4. Ce 3d XPS spectra of different gold-ceria samples before and after the CO oxidation reaction.

Reviewer #2:

A. This manuscript details the study of different gold samples (ionic salts, clusters and nanoparticles) on ceria for the oxidation of Carbon monoxide using in situ X-ray absorption spectroscopy and STEM to understand speciation of the Au catalysts. Results show that metallic gold species are present in samples having high catalytic activity. DRIFTS spectroscopy was also used to probe the nature of the Au species for each of the samples.

B. This is an interesting paper that has, for the most part, well validated conclusions from experimental work. The X-ray absorption work is very well done. As the authors note in the manuscript well, there have been a good number of other attempts to understand Au oxidation catalysts via X-ray absorption spectroscopy, and it is fair to say that those results have been all over the map in terms of identifying the catalytically active species. This is, in part, due to the difficulty in making samples that have distinctive single phases, but is also likely due both to X-ray beam damage of samples during acquisition and the fact that X-ray absorption spectroscopy is a bulk technique that is not well suited for identifying minor species. Nevertheless, I think the authors have done well to make three distinctive samples, and have commented on how they avoided beam damage in the samples, thus I think there is novelty in the final conclusions of this work, and it should be acceptable for publication in Nature Communications pending minor revisions.

C. Quality of data looks excellent for X-ray absorption work. It would be useful if the authors could provide higher magnification images of the STEM images in Figure 1, particularly for the atomic gold samples.

D. Statistical treatment of X-ray absorption work is very good, and uncertainties are reported.

E. The conclusions of the paper are well validated from experimental results.

F. Suggested improvements:

1. Improve magnification of STEM images (see point C).

2. Improve grammar throughout - currently the grammar in the paper is quite poor and detracts from the manuscript substantially.

G. References are appropriate, and I think previous X-ray absorption work on such systems is fairly well cited.

H. Clarity needs work - as mentioned in section F, the grammar in the manuscript submitted is quite weak, and could be improved significantly. This would greatly improve the readability of the paper.

Response: We highly appreciate the reviewer. In the revised manuscript, we have improved the magnification of STEM images for Au_atom and Au_cluster in **Figure 1**. Meanwhile, we have carefully corrected the grammar mistakes and enhanced the writing of this paper.

Reviewer #3:

Gold nanocatalyst is famous for its extraordinarily high activity for low temperature CO oxidation. However, the nature of active sites has long been disputed, for example, cationic or metallic, single atoms or clusters or nanoparticles. The present work by Prof Jia and co-workers appears to reveal the tip of the iceberg. They prepared gold single atoms, clusters (<2 nm), and nanoparticles (3-4 nm) on a ceria nanorod support, and compared their activities for RT CO oxidation meanwhile detected the local structure change and oxidation state change of different gold species during the reaction. Based on these in-situ characterization results, they conclude that gold metallic nanoparticles are much more active than cationic gold single atoms for RT CO oxidation. I enjoyed the application of the in situ XAFS and dynamic adsorption-desorption IR techniques, which provide reliable evidence to support the conclusion. Nevertheless, the catalysis by gold is critically correlated with the support nature, which is not addressed in the work.

1. The authors claim that the three catalysts have the same support. However, the difference in preparation history of the catalysts may cause the change of the support surface properties. For example, single atom catalyst was calcined at 400 °C while the cluster catalyst was further reduced at 300 °C. The reduction treatment will bring about the change of surface lattice oxygen of CeO₂ support in addition to the aggregation of gold single atoms. CO titration in combination of XPS may help to address this concern.

Response: We genuinely thank this reviewer for his/her constructive suggestions. To further investigate the structure of cerium oxide support during synthesis, especially the gold-deposition process, we have included new experimental data on O₂-TPD (Figure L3), XPS of Ce 3*d* (Figure S3 or Figure L4 here) and CO titration (Figure S12 or Figure L1 here) in the revised manuscript, and added the related discussion in the text (Page 4, Line 17–19).

No O₂ (*m/z* = 32) signals were detected up to 800 °C in O₂-TPD for all the gold-ceria catalysts (Figure L3), revealing the negligible amount of adsorbed oxygen species. Meanwhile, the corresponding XPS spectra before and after the CO oxidation reaction confirmed the almost identical profiles for Ce 3*d*, indicating the similar ratios of Ce³⁺/Ce⁴⁺ for

ceria nanorods between the investigated samples (Figure S3 or Figure L4 here). Therefore, the surface of ceria support in this work has the same structure, in spite of the different post-treatment conditions (400 °C, air; 300 °C, 5% H₂/He) during the gold deposition process.

On the other hand, according to the CO titration data (Figure S12 or Figure L1 here), we found that Au_{atom} and pure ceria support have zero CO₂ formation at room temperature, due to the weak adsorption of CO by ionic gold species or the absence of activated oxygen; while Au_{cluster} and Au_{particle} have significant amount of CO₂ produced at room temperature, active oxygen species, because of the strong adsorption of CO by metallic gold species and the presence of activated oxygen. Thus, the oxygen species at the interfaces between gold and ceria could be altered after the addition of different gold species of single atoms, clusters or particles. These phenomena were well consistent with the *in-situ* DRIFTS results (Figure 5). The corresponding revision is seen the revised text (Page 12, Line 5–10).

Figure L3. O₂ evolution ($m/z = 32$) during O₂-TPD over gold-ceria samples. The O₂-TPD experiments were performed at Builder PCSA-1000 System equipped with a mass spectrometer (AMETEK, DYCOR LC-D200). First, 100 mg sample powders were pretreated at 300 °C (10 °C/min) in air (50 mL/min) for 30 min and in addition subsequent pretreatment (300 °C, 5% H₂/He, 30 min) was required for Au_{cluster}. After cooling down, the measured sample was continued to be purged with pure O₂ (50 mL/min) at room temperature for 1 h to obtain saturated adsorption. Then, the feed gas was switched to pure He (30 mL/min) at room temperature until the stabilization of baseline. The O₂-TPD process was done in pure He (30

mL/min) from room temperature to 800 °C (10 °C/min) with the simultaneous collection on O₂ signals ($m/z = 32$).

Figure L4. Ce 3d XPS spectra of different gold-ceria samples before and after the CO oxidation reaction.

Figure L1. CO₂ ($m/z = 44$) evolution collected during CO titration over gold-ceria samples at room temperature.

2. The HAADF-STEM images seem to show that the ceria nanorods have a large number of voids, which means that it has defect-rich surface. Do these defects affect the catalysis by gold?

Response: We totally agree with this reviewer. During the various types of ceria supports, the presence of voids is typical for nanorods. This may be caused by the dehydration process during the growth of rod-like CeO₂ nanocrystals,¹⁴ and could generate more surface defects, e.g. oxygen vacancies¹⁵. It was also reported that such oxygen vacancies are required to stabilize active gold species at the Au-CeO₂ interfaces.¹⁶ Therefore, we believe that the formation of defects is very important to affect the gold catalysis in this work.

3. It is difficult to understand the models in Fig. 3c. Where do the oxygen atoms come from? The authors show two Au-O shells in single-atom catalyst but they claim that no any Au-Ce distance from Au-O-Ce structure. Does this mean gold single atoms do not coordinate with ceria via oxygen atoms? If it is true, how is it stabilized by the support?

Response: This reviewer raised a very good point. Actually, we have tried different models such as Au-O (1st shell) + Au-Ce (2nd shell), other than the claimed two shells of Au-O, to fit the EXAFS data for the as-prepared Au_{atom} sample (Figure L5 and Table L1). However, reasonable results were displayed for the two shells of Au-O model only. This conclusion also agrees with the reported EXAFS fitting results by Deng et. al. in Au-CeO₂ system¹⁷.

Figure L5. EXAFS fitting results on the as-prepared Au_{atom} sample via the two shells of Au-O (red, solid) model and the Au-O (1st shell) + Au-Ce (2nd shell) model (red, dot).

Here, we think the missing of Au-Ce shell does not mean the absence of Au-O-Ce interaction at the interfaces between gold and ceria support. As an X-ray based technique, XAFS is still more sensitive to the ordering models, although it may catch some less ordering structures. In this case, the Au-O shell at the longer distance (~ 3.5 Å) is more ordering than the Au-Ce shell located around 3.3 Å in the form of Au-O-Ce interaction. Thus, we failed to determine any Au-Ce component by the EXAFS fit. Thus, we have emphasized the strong interaction between gold single atoms and surface oxygen species by the analysis of EXAFS data in the text (Page 8, Line 10-13).

Table L1. EXAFS fitting results on distance (R) and coordination number (CN) of Au-O and Au-Au shells for the as-prepared Au_atom sample by different models.

Sample	Au-O		Au-Ce	
	R (Å)	CN	R (Å)	CN
Two shells of Au-O	1.96±0.01	3.0±0.3	–	–
	3.49±0.03	5.7±1.5		
Au-O (1 st shell) + Au-Ce (2 nd shell)	1.96±0.01	4.2±0.4	3.28±0.02	2.8±0.9

4. In gold single-atom catalyst, 1.2 wt% Au loading is relatively high for atomically dispersion. I guess the ceria nanorod support has many defects that can stabilize single atoms of gold, is it right?

Response: This is a good point and we agree with this reviewer. It has been reported that the {110} surface of ceria nanorods has the lower formation energy of oxygen vacancy¹⁸ and thus can stabilize more active gold species during synthesis as discussed as above (Question 2). Recently, Qiao et. al. prepared good single atoms via coprecipitation with ceria (Au₁/CeO₂) up to 0.3 wt.% Au, which exhibited super reactivity for the preferential oxidation of CO.¹⁹ In this work, the further increase of single-atom gold concentration is related to the use of ceria nanorods. We have added the above discussion in the revised manuscript (Page 8, Line 27; Page 9, Line 1–3).

Reference:

- (1) Wang, G. Y.; Mei, D. H.; Glezakou, V. A.; Li, J.; Rousseau, R. *Nat. Commun.* **2015**, *6*, 6511, DOI: 10.1038/ncomms7511.
- (2) Yu, Y. B.; Takei, T.; Ohashi, H.; He, H.; Zhang, X. L.; Haruta, M. *J. Catal.* **2009**, *267*, 121–128.
- (3) Pal, R.; Wang, L. M.; Pei, Y.; Wang, L. S.; Zeng, X. C. *J. Am. Chem. Soc.* **2012**, *134*, 9438–9445.
- (4) Wu, Z. L.; Li, M.; Overbury, S. H. *J. Catal.* **2012**, *285*, 61–73.
- (5) Widmann, D.; Behm, R. *J. Accounts Chem. Res.* **2014**, *47*, 740–749.
- (6) Jia, C. Y.; Zhang, G. Z.; Zhong, W. H.; Jiang, J. *ACS Appl. Mater. Inter.* **2016**, *8*, 10315–10323.
- (7) Qiao, B. T.; Liu, J. X.; Wang, Y. G.; Lin, Q. Q.; Liu, X. Y.; Wang, A. Q.; Li, J.; Zhang, T.; Liu, J. Y. *ACS Catal.* **2015**, *5*, 6249–6254.
- (8) Wu, Z. L.; Jiang, D. E.; Mann, A. K. P.; Mullins, D. R.; Qiao, Z. A.; Allard, L. F.; Zeng, C. J.; Jin, R. C.; Overbury, S. H. *J. Am. Chem. Soc.* **2014**, *136*, 6111–6122.
- (9) Schalow, T.; Brandt, B.; Laurin, M.; Schauermaun, S.; Libuda, J.; Freund, H.-J. *J. Catal.* **2009**, *267*, 121–128.
- (10) Guzman, J., Carrettin, S.; Corma, A. *J. Am. Chem. Soc.* **2005**, *127*, 3286–3287.
- (11) Denkwitz, Y.; Zhao, Z.; Hormann, U.; Kaiser, U.; Plzak, V.; Behm, R. *J. Catal.* **2007**, *251*, 363–373.
- (12) Katsaounis, A.; Nikopoulou, Z.; Verykios, X. E.; Vayenas, C. G. *J. Catal.* **2004**, *226*, 197–209.
- (13) You, R.; Zhang, Y. X.; Liu, D. S.; Meng, M.; Zheng, L. Y.; Zhang, J.; Hu, T. D. *J. Phys. Chem. C* **2014**, *118*, 25403–25420.
- (14) Mai, H. X.; Sun, L. D.; Zhang, Y. W.; Si, R.; Feng, W.; Zhang, H. P.; Liu, H. C.; Yan, C. H. *J. Phys. Chem. C* **2005**, *109*, 24380–24385.
- (15) Si, R.; Flytzani-Stephanopoulos, M. *Angew. Chem. Int. Ed.* **2008**, *47*, 2884–2887.
- (16) Liu, Z.-P.; Jenkins, S. J.; King, D. A. *Phys. Rev. Lett.* **2005**, *94*, 196102.
- (17) Deng, W.-L.; Frenkel, A. I.; Si, R.; Flytzani-Stephanopoulos, M. *J. Phys. Chem. C* **2008**, *112*, 12834–12840.
- (18) Sayle, T. X. T.; Parker, S. C.; Sayle, D. C. *Phys. Chem. Chem. Phys.* **2005**, *7*, 2936–2941.
- (19) Qiao, B.-T.; Liu, J.-X.; Wang, Y.-G.; Lin, Q.-Q.; Liu, X.-Y.; Wang, A.-Q.; Li, J.; Zhang, T.; Liu, J.-Y. *ACS Catal.* **2015**, *5*, 6249–6254.

REVIEWERS' COMMENTS:

Reviewer #1 (Remarks to the Author):

The authors have answered in a very detail mode all the question done by the reviewers. The manuscript is suitable for publication.

Reviewer #3 (Remarks to the Author):

Most of my concerns have been addressed satisfactorily, so I recommend the revised manuscript be published in Nat. Commun.

Responses to the reviewer's comments

Reviewer #1:

Comment: The authors have answered in a very detail mode all the questions done by the reviewers. The manuscript is suitable for publication.

Response: Thanks for the reviewer's comment. We are very glad that our responses have been approved by the reviewer.

Reviewer #3:

Comment: Most of my concerns have been addressed satisfactorily, so I recommend the revised manuscript be published in *Nat. Commun.*

Response: Thanks for the reviewer's recommendation.